# BLACK-BOX ATTACKS ON DEEP NEURAL NETWORKS VIA GRADIENT ESTIMATION

**Arjun Nitin Bhagoji** *
Department of Electrical Engineering
Princeton University

**Warren He, Bo Li & Dawn Song**
EECS Department
University of California, Berkeley

## ABSTRACT

In this paper, we propose novel Gradient Estimation black-box attacks to generate adversarial examples with query access to the target model's class probabilities, which do not rely on transferability. We also propose strategies to decouple the number of queries required to generate each adversarial example from the dimensionality of the input. An iterative variant of our attack achieves close to 100% attack success rates for both targeted and untargeted attacks on DNNs. We show that the proposed Gradient Estimation attacks outperform all other black-box attacks we tested on both MNIST and CIFAR-10 datasets, achieving attack success rates similar to well known, state-of-the-art white-box attacks. We also apply the Gradient Estimation attacks successfully against a real-world content moderation classifier hosted by Clarifai.

## 1 INTRODUCTION

The ubiquity of machine learning provides adversaries with both opportunities and incentives to develop strategic approaches to fool learning systems and achieve their malicious goals. Many attack strategies devised so far to generate adversarial examples to fool learning systems have been in the white-box setting, where adversaries are assumed to have access to the learning model (Szegedy et al., 2014; Goodfellow et al., 2015; Carlini & Wagner, 2017; Kurakin et al., 2016). However, in many realistic settings, adversaries may only have black-box access to the model, i.e. they have no knowledge about the details of the learning system such as its parameters, but they may have query access to the model's predictions on input samples, including class probabilities. Most existing black-box attacks on DNNs have relied on *transferability* (Papernot et al., 2016; Moosavi-Dezfooli et al., 2016; Papernot et al., 2017), where adversarial examples crafted for a local surrogate model can be used to attack the target model.

**New black-box attacks.** In this short paper, we provide an overview of our powerful new black-box *Gradient Estimation* attacks on DNNs using *limited query access* which achieve attack success rates close to that of white-box attacks and do not rely on transferability. These attacks do not need any access to a representative dataset or any knowledge of the target model architecture. In the Gradient Estimation attacks, the adversary adds perturbations proportional to the *estimated gradient*, instead of the true gradient as in white-box attacks. More details and results can be found in our technical report (Bhagoji et al., 2017). The code to reproduce our results can be found at `https://github.com/sunblaze-ucb/blackbox-attacks`.

**Query-reduction strategies.** In order to reduce the number of queries needed to the target model, we propose two novel strategies: *random feature grouping* and *principal component analysis (PCA) based query reduction*. In our experiments with the Gradient Estimation attacks on state-of-the-art models on MNIST (784 dimensions) and CIFAR-10 (3072 dimensions) datasets, we find that they match white-box attack performance, achieving attack success rates up to 90% for single-step attacks in the untargeted case and up to 100% for iterative attacks in both targeted and untargeted cases (Tables 1 and 2). We achieve this performance with just 200 to 800 queries per sample for single-step attacks and around 8,000 queries for iterative attacks.

---

*Work done while visiting UC Berkeley

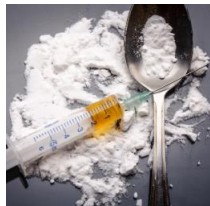 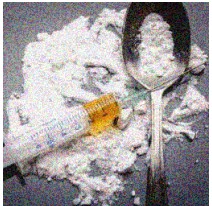

Figure 1: Sample adversarial images of Gradient Estimation attacks on Clarifai's Content Moderation model. **Left**: original image, classified as 'drug' with a confidence of 0.99. **Right**: adversarial sample with $\epsilon = 32$, classified as 'safe' with a confidence of 0.96.

**Attacking real-world systems.** To demonstrate the effectiveness of our Gradient Estimation attacks in the real world, we also carry out a practical black-box attack using these methods against the Not Safe For Work (NSFW) classification and Content Moderation models developed by Clarifai, which we choose due to their socially relevant application. We have demonstrated successful attacks (Figure 1) with just around 200 queries per image, taking around a minute per image. The full set of images can be found at `https://sunblaze-ucb.github.io/blackbox-attacks/`

**Related Work.** To the best of our knowledge, the only previous literature on query-based black-box attacks in the deep learning setting is independent work by Narodytska & Kasiviswanathan (2016) and Chen et al. (2017). Narodytska & Kasiviswanathan (2016) propose a greedy local search to generate adversarial examples by perturbing randomly chosen pixels and using those which have a large impact on the output probabilities. Chen et al. (2017) propose a black-box attack method named ZOO, which also uses the method of finite differences to estimate the derivative of a function. However, while we propose attacks that compute an adversarial perturbation, approximating FGSM and iterative FGS; ZOO approximates the Adam optimizer, while trying to perform coordinate descent on the loss function proposed by Carlini & Wagner (2017).

## 2 QUERY BASED ATTACKS: GRADIENT ESTIMATION ATTACK

In this section, we present our novel Gradient Estimation based attacks in the *targeted* setting. Attacks in the untargeted setting are simple extensions which can be found in our technical report (Bhagoji et al., 2017).

**Threat model and justification**: We assume that the adversary can obtain the vector of output probabilities for any input $\mathbf{x}$. The set of queries the adversary can make is then $\mathcal{Q}_f = \{\mathbf{p}^f(\mathbf{x}), \forall \mathbf{x}\}$. Note that an adversary with access to the softmax probabilities will be able to recover the logits (outputs of the penultimate layer) up to an additive constant, by taking the logarithm of the softmax probabilities. The logits are represented as a vector $\boldsymbol{\phi}^f(\mathbf{x}) \in \mathbb{R}^{|\mathcal{Y}|}$.

Let the function whose gradient is being estimated be $g(\mathbf{x})$. The input to the function is a $d$-dimensional vector $\mathbf{x}$, whose elements are represented as $\mathbf{x}_i$, where $i \in [1, \ldots, d]$. The canonical basis vectors are represented as $\mathbf{e}_i$, where $\mathbf{e}_i$ is 1 only in the $i^{th}$ component and 0 everywhere else. Then, a two-sided finite difference estimation of the gradient of $g$ with respect to $\mathbf{x}$ is given by

$$\mathrm{FD}_{\mathbf{x}}(g(\mathbf{x}), \delta) = \begin{bmatrix} \frac{g(\mathbf{x}+\delta\mathbf{e}_1)-g(\mathbf{x}-\delta\mathbf{e}_1)}{2\delta} \\ \vdots \\ \frac{g(\mathbf{x}+\delta\mathbf{e}_d)-g(\mathbf{x}-\delta\mathbf{e}_d)}{2\delta} \end{bmatrix}. \tag{1}$$

The logit loss (Carlini & Wagner, 2017) of a network $f$ at an input $\mathbf{x}$ with respect to a target $T$ is $\ell(\mathbf{x}, y) = \max\{\boldsymbol{\phi}^f(\mathbf{x})_i : i \neq T\} - \boldsymbol{\phi}^f(\mathbf{x})_T$. White-box adversarial examples based on this loss are $\mathbf{x}_{\mathrm{adv}} = \mathbf{x} - \epsilon \cdot \mathrm{sign}(\nabla_{\mathbf{x}}(\max(\boldsymbol{\phi}(\mathbf{x})_i : i \neq T) - \boldsymbol{\phi}(\mathbf{x})_T))$. Since an adversary with query access to the softmax probabilities can estimate the logits up to an additive constant which cancels out for the logit loss, she just has to plug in the finite difference estimate of the gradient of the logit loss to obtain an adversarial sample: $\mathbf{x}_{\mathrm{adv}} = \mathbf{x} - \epsilon \cdot \mathrm{sign}(\mathrm{FD}_{\mathbf{x}}(\max(\boldsymbol{\phi}(\mathbf{x})_i : i \neq T) - \boldsymbol{\phi}(\mathbf{x})_T, \delta))$. An iterative attack with step size $\alpha$, $\mathcal{H}$ as the constraint set and $t + 1$ iterations using the logit loss is:

$$\mathbf{x}_{\mathrm{adv}}^{t+1} = \Pi_{\mathcal{H}}\left(\mathbf{x}_{\mathrm{adv}}^t - \alpha \cdot \mathrm{sign}(\mathrm{FD}_{\mathbf{x}_{\mathrm{adv}}^t}(\max(\boldsymbol{\phi}(\mathbf{x}_{\mathrm{adv}}^t)_i : i \neq T) - \boldsymbol{\phi}(\mathbf{x}_{\mathrm{adv}}^t)_T, \delta))\right). \tag{2}$$

## 2.1 QUERY REDUCTION

The major drawback of the approximation based black-box attacks is that the number of queries needed per adversarial sample is large. For an input with dimension $d$, the number of queries will be exactly $2d$ for a two-sided approximation. This may be too large when the input is high-dimensional.

**Random grouping.** The simplest way to group features is to choose, without replacement, a random set of features. The gradient can then be simultaneously estimated for all these features. If the size of the set chosen is $k$, then the number of queries the adversary has to make is $\lceil \frac{d}{k} \rceil$. When $k = 1$, this reduces to the case where the partial derivative with respect to every feature is found, as in the previous section. Thus, the quantity being estimated is not the gradient itself, but an index-wise averaged version of it.

**Query reduction using PCA components.** A more principled way to reduce the number of queries the adversary has to make to estimate the gradient is to compute directional derivatives (Hildebrand, 1962) along the principal components as determined by principal component analysis (PCA) (Shlens, 2014), which requires the adversary to have access to a set of data which is representative of the training data. The estimate of the gradient used is then an approximation of the true gradient by the sum of its projection along the top $k$ principal components.

## 3 EXPERIMENTAL RESULTS

Our empirical evaluation is on state-of-the-art neural networks on the MNIST (LeCun & Cortes, 1998) and CIFAR-10 (Krizhevsky & Hinton, 2009) datasets. The architecture and training details for all models are given in Appendix A.1. We use the logit loss (abbreviated as logit). In all of these attacks, the adversary's perturbation is constrained using the $L_\infty$ distance. The attack success rate, is the fraction of examples that meets the adversary's goal: $f(\mathbf{x}_{\mathrm{adv}}) \neq y$ for untargeted attacks and $f(\mathbf{x}_{\mathrm{adv}}) = T$ for targeted attacks with target $T$. For the Iterative Gradient Estimation attacks, we use $\alpha = 0.01$ and $t = 40$ for MNIST and $\alpha = 1.0$ and $t = 10$ for CIFAR-10 throughout. Table 3 (Appendix) contains a brief summary of all the attacks we evaluated. We show some examples of successful untargeted adversarial examples in Figure 2.

| MNIST | White-box | | Gradient Estimation (FD) | | Gradient Estimation (Query Reduction) | | | |
|---|---|---|---|---|---|---|---|---|
| Model | Single-step FGS (logit) | Iterative IFGS (logit) | Single-step [1568] FD-logit | Iterative [62720] IFD-logit | Single-step [$\sim 200$] PCA-100 | RG-8 | Iterative [8000] PCA-100 | RG-8 |
| A | 30.1 (6.1) | 99.6 (2.7) | 29.9 (6.1) | **99.7 (2.7)** | 23.2 (5.9) | 15.9 (5.9) | 96.2 (3.3) | 73.8 (2.5) |
| B | 29.6 (6.2) | **98.7 (2.4)** | 29.3 (6.3) | **98.7 (2.4)** | 29.0 (6.3) | 17.8 (6.3) | 93.9 (2.9) | 73.7 (2.6) |
| CIFAR-10 | White-box | | Gradient Estimation (FD) | | Gradient Estimation (Query Reduction) | | | |
| Model | Single-step FGS (logit) | Iterative IFGS (logit) | Single-step [6144] FD-logit | Iterative [61440] IFD-logit | Single-step [$\sim 800$] PCA-400 | RG-8 | Iterative [8000] PCA-400 | RG-8 |
| Resnet-32 | 23.5 (436.0) | 100.0 (89.5) | 23.0 (437.0) | 100.0 (89.5) | 21.0 (438.2) | 19.0 (438.1) | 81.0 (222.8) | 97.0 (126.1) |
| Resnet-28-10 | 27.6 (436.5) | 100.0 (99.0) | 28.0 (436.1) | 100.0 (98.3) | 23.0 (433.7) | 20.0 (433.7) | 72.0 (253.1) | 94.0 (132.4) |

Table 1: **Targeted black-box attacks**: attack success rates. The number in parentheses () for each entry is $\Delta(\mathbf{X}, \mathbf{X}_{\mathrm{adv}})$, the average $L_2$ distortion over all examples used in the attack. This is used to compare the stealth of different attacks. The number in brackets [] beside the Single-step and Iterative descriptors gives the number of queries needed for each type of attack. The per-pixel perturbation limits are $\epsilon = 0.3$ for MNIST (**Top**) and $\epsilon = 8$ for CIFAR-10 (**Bottom**).

**Targeted attack success**. From Table 1, it is clear that the Gradient Estimation attacks *match attack success rates for white-box attacks in the targeted case*. Further, the query reduction methods reduce the number of queries greatly while maintaining high attack success rates.

**Untargeted attacks and other query-based attacks**. In Table 2 in the Appendix, attack success rates for untargeted Gradient Estimation attacks are given and compared with other query-based attacks for completeness. We experimented with Particle Swarm Optimization (PSO),[1] a commonly used evolutionary optimization strategy, to construct adversarial examples, but found it to be prohibitively slow even for MNIST. We also used the Simultaneous Perturbation Stochastic Approximation (SPSA) method, which is similar to the method of Finite Differences, but estimates the gradient of the loss along a *random direction* $\mathbf{r}$ at each step, instead of along the canonical basis vectors. We find that the Gradient Estimation attack outperforms the other black-box attacks.

---

[1]Using freely available code from `http://pythonhosted.org/pyswarm/`

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

# A APPENDIX

## A.1 MODEL TRAINING DETAILS

In this section, we present the architectures and training details for models trained on both the MNIST and CIFAR-10 datasets.

**MNIST.** The model details for the 2 models trained on the MNIST dataset are as follows:

1. Model A (3,382,346 parameters): Conv(64, 5, 5) + Relu, Conv(64, 5, 5) + Relu, Dropout(0.25), FC(128) + Relu, Dropout(0.5), FC + Softmax
2. Model B (710,218 parameters) - Dropout(0.2), Conv(64, 8, 8) + Relu, Conv(128, 6, 6) + Relu, Conv(128, 5, 5) + Relu, Dropout(0.5), FC + Softmax

Model A has both convolutional layers as well as fully connected layers. Model B, on the other hand, does not have fully connected layers and has an order of magnitude fewer parameters.

Models A and B achieve greater than 99% classification accuracy on the test data.

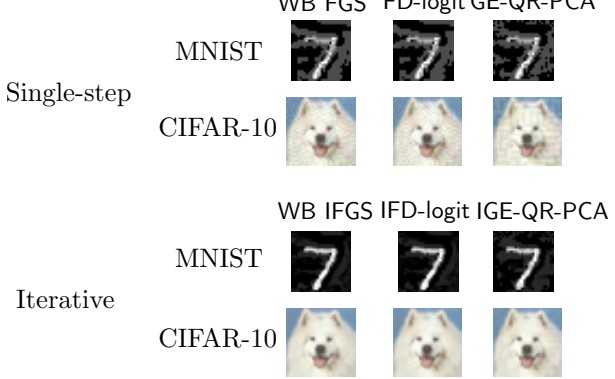

Figure 2: Untargeted **adversarial examples** on Model A on MNIST and Resnet-32 on CIFAR-10. All attacks use the logit loss. Perturbations in the images generated using single-step attacks are far smaller than those for iterative attacks. The '7' from MNIST is classified as a '3' by all single-step attacks and as a '9' by all iterative attacks. The *dog* from CIFAR-10 is classified as a *bird* by the white-box FGS and Finite Difference attack, and as a *frog* by the Gradient Estimation attack with query reduction.

| Query-based attack | Attack success (Distortion) | No. of queries | Time per sample (s) |
|---|---|---|---|
| Finite Diff. | 92.9 (6.1) | 1568 | $8.8 \times 10^{-2}$ |
| Gradient Estimation (RG-8) | 61.5 (6.0) | 196 | $1.1 \times 10^{-2}$ |
| Iter. Finite Diff. | **100.0 (2.1)** | 62720 | 3.5 |
| Iter. Gradient Estimation (RG-8) | 98.4 (1.9) | 8000 | 0.43 |
| Particle Swarm Optimization | 84.1 (5.3) | 10000 | 21.2 |
| SPSA | 96.7 (3.9) | 8000 | 1.25 |

Table 2: Comparison of **untargeted query-based black-box attack** methods. All results are for attacks using the first 1000 samples from the MNIST dataset on Model A and with an $L_\infty$ constraint of 0.3. The logit loss is used for all methods expect PSO, which uses the class probabilities.

**CIFAR-10.** As their name indicates, Resnet-32 and Resnet-28-10 are ResNet variants (He et al., 2016; Zagoruyko & Komodakis, 2016). In particular, Resnet-32 is a standard 32 layer ResNet with no width expansion, and Resnet-28-10 is a wide ResNet with 28 layers with the width set to 10, based on the best performing ResNet from Zagoruyko & Komodakis (TensorFlow Authors). The width indicates the multiplicative factor by which the number of filters in each residual layer is increased.

| | Attack | | | | Abbreviation | Untargeted | Targeted |
|---|---|---|---|---|---|---|---|
| **Black-box** | Finite-difference gradient estimation | | Steps | Loss function | | | |
| | | | Single-step | Cross-entropy | FD-xent | + | + |
| | | | | Logit-based | FD-logit | + | + |
| | | | Iterative | Cross-entropy | IFD-xent | + | + |
| | | | | Logit-based | IFD-logit | + | + |
| | Query-reduced gradient estimation | Technique | Steps | Loss function | | | |
| | | Random grouping | Single-step | Logit-based | GE-QR (RG-$k$, logit) | + | + |
| | | | Iterative | Logit-based | IGE-QR (RG-$k$, logit) | + | + |
| | | PCA | Single-step | Logit-based | GE-QR (PCA-$k$, logit) | + | + |
| | | | Iterative | Logit-based | IGE-QR (PCA-$k$, logit) | + | + |
| **White-box** | Fast gradient sign (FGS) | | Steps | Loss function | | | |
| | | | Single-step | Cross-entropy | WB FGS-xent | + | + |
| | | | | Logit-based | WB FGS-logit | + | + |
| | | | Iterative | Cross-entropy | WB IFGS-xent | + | + |
| | | | | Logit-based | WB IFGS-logit | + | + |

Table 3: Attacks evaluated in this paper

For each model architecture, we train 3 models, one on only the CIFAR-10 training data, one using standard adversarial training and one using ensemble adversarial training. Resnet-32 is trained for 125,000 steps, Resnet-28-10 is trained for 167,000 steps and Std.-CNN is trained for 100,000 steps on the benign training data. All models were trained with a batch size of 128. The two ResNets achieve close to state-of-the-art accuracy (Benenson) on the CIFAR-10 test set, with Resnet-32 at 92.4% and Resnet-28-10 at 94.4%.

