# OpenReview forum: "Black-box Attacks on Deep Neural Networks via Gradient Estimation"
_ICLR.cc/2018/Workshop — Accept_

### Official Review · AnonReviewer2 · 2018-03-05
**An interesting paper**

**Rating:** 7
**Confidence:** 4

**Review:**

The paper tackles adversarial example generation when the model that produces predictions is treated as a black box, with only information it provides is the class probabilities. Authors' approach also does not rely on building a surrogate model based on the predictions from the real model, and then finding the adversarial examples for this surrogate model.
The gist is that the adversary adds perturbations proportional to the estimated gradient, which can be obtained by querying the probabilities from the black box model 2d times, where d is the dimension of the input.  Authors also suggest two extensions that allow to reduce the number of queries needed

Overall the paper is relatively easy to read, however some background information is missing. For example, targeted vs untargeted setting, single vs iterative. A couple of sentences explaining this would have gone the long way.

- What is f (in the set of queries the adversary can make)
- H (constraint set) is also not introduced - what does it represent. Is it the max change that can be introduced on a input instance as not to change it visually. Why and how does it participate in formula (2)
- Random groupping is not well explained either.

For mnist and cifar experiments, how were those learning rates chosen (eg how many tries you had to perform to choose those params - these should be included into the "number of iterations")

---

### Official Review · AnonReviewer3 · 2018-03-07
**An interesting paper on black-box attack**

**Rating:** 7
**Confidence:** 4

**Review:**

This paper proposes a black-box gradient estimation attacks on DNNs using limited query access. It achieves attack success rates close to that of white-box attacks and do not rely on transferability. This short paper is very well written and is easy to follow. It is an interesting paper.

I have two comments.

(1) In the experiments shown in Table 1, the proposed method is only compared with the white-box attacks. Given that the proposed method is highly related to another black-box attack method ZOO proposed in Chen et al. (2017), the authors are suggested to add ZOO method in their numerical comparison.

(2) The authors mentioned that the tuning parameters were chosen as \alpha = 0.01 and t = 40 for MNIST and \alpha = 1.0 and t = 10 for CIFAR-10. Can the authors provide some explanations on how to select these parameters in practice? Is there any data-driven way to choose them?

---

### Official Review · AnonReviewer1 · 2018-03-09
**Promising approach  to the generation of adversarial attacks through gradient estimation in a black box setting.**

**Rating:** 6
**Confidence:** 3

**Review:**

This work proposes a  black box approach  to the generation of adversarial attacks through gradient estimation.
The proposed method is designed in a rather realistic thus difficult setting:
  - Black box (no attacked model prior knowledge),
  - No surrogate model (transferability) used to develop the adversarial attacks.
  - Query based attacks with 2 query reduction approaches (to alleviate black-box approaches' need for large numbers of queries).
it does require access to attacked model's:
  - Outputs to supplied inputs and
  - class probabilities.
The approach is clear and well presented, through the limitations of a short paper.
The experimental part presents interesting results when compared to white box approaches especially with query reduction.
Appreciated is the fact that the evaluations were conducted on both MNIST handwritten digits and CIFAR-10 image datasets which provide for varied types of images.

---

### Decision · Program_Chairs · 2018-03-20
**ICLR 2018 Workshop Acceptance Decision**

**Decision:**

Accept

**Comment:**

Congratulations, your paper was accepted to the ICLR workshop.